# SODA10M: Towards Large-Scale Object Detection Benchmark for Autonomous Driving

**Jianhua Han**[1*]    **Xiwen Liang**[2*]    **Hang Xu**[1†]    **Kai Chen**[3]    **Lanqing Hong**[1]

**Chaoqiang Ye**[1]    **Wei Zhang**[1]    **Zhenguo Li**[1]    **Xiaodan Liang**[2†]    **Chunjing Xu**[1]

## Abstract

Aiming at facilitating a real-world, ever-evolving and scalable autonomous driving system, we present a large-scale benchmark for standardizing the evaluation of different self-supervised and semi-supervised approaches by learning from raw data, which is the first and largest benchmark to date. Existing autonomous driving systems heavily rely on 'perfect' visual perception models (e.g., detection) trained using extensive annotated data to ensure the safety. However, it is unrealistic to elaborately label instances of all scenarios and circumstances (e.g., night, extreme weather, cities) when deploying a robust autonomous driving system. Motivated by recent powerful advances of self-supervised and semi-supervised learning, a promising direction is to learn a robust detection model by collaboratively exploiting large-scale unlabeled data and few labeled data. Existing dataset (e.g., KITTI, Waymo) either provides only a small amount of data or covers limited domains with full annotation, hindering the exploration of large-scale pre-trained models. Here, we release a Large-Scale Object Detection benchmark for Autonomous driving, named as **SODA10M**, containing 10 million unlabeled images and 20K images labeled with 6 representative object categories. To improve diversity, the images are collected every ten seconds per frame within 32 different cities under different weather conditions, periods and location scenes. We provide extensive experiments and deep analyses of existing supervised state-of-the-art detection models, popular self-supervised and semi-supervised approaches, and some insights about how to develop future models. We show that SODA10M can serve as a promising pre-training dataset for different self-supervised learning methods, which gives superior performance when finetuning autonomous driving downstream tasks. This benchmark will be used to hold the ICCV2021 SSLAD challenge. The data and more up-to-date information have been released at https://soda-2d.github.io.

## 1    Introduction

Autonomous driving technology has been significantly accelerated in recent years because of its great potential in reducing accidents, saving human lives and improving efficiency. As an essential module in the visual perception system, object detection in road images plays one of the most critical roles for autonomous driving.

Performances of current object detection approaches, however, may be limited by the currently available datasets [7, 64, 49], due to the drawbacks of existing benchmarks. First, the diversity of data

---

[1] Huawei Noah's Ark Lab        [2] Sun Yat-Sen University
[3] Hong Kong University of Science and Technology    * These two authors contribute equally.
† Corresponding authors: xdliang328@gmail.com & xu.hang@huawei.com

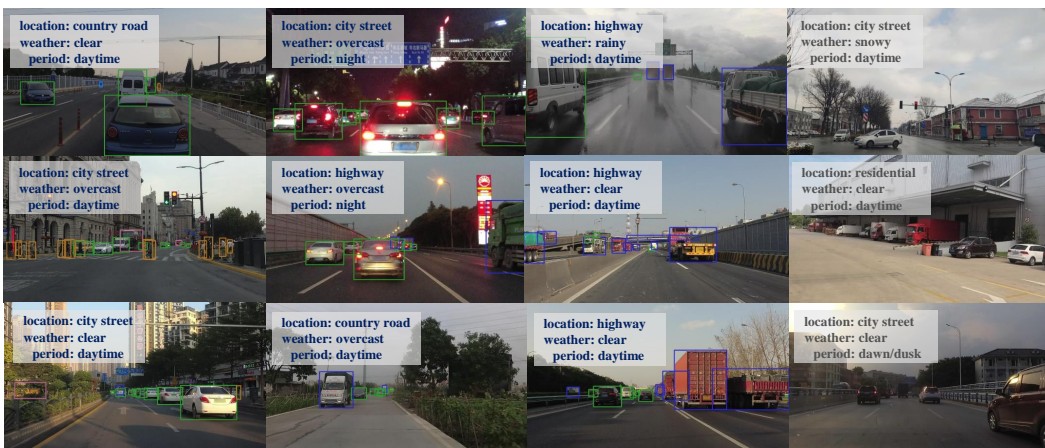

Figure 1: Examples of challenging environments in our dataset. The first three columns of images are from SODA10M labeled set, and the last column is from the unlabeled set. Our dataset includes a diverse set of 10 million images under different weather conditions, periods and locations.

sources is lacking. For example, the largest self-driving dataset in existence, Waymo Open [49], was collected from only three cities, covering only a few scenarios and circumstances. Models trained on these datasets may overfit to specific scenarios or characteristics. Second, existing datasets are usually fully annotated but limited in scale due to the cost of data annotation. They are not able to support the exploration of autonomous driving with huge volumes of unlabeled data.

Numerous self-supervised techniques [4, 19, 21, 5] have been developed for vision tasks to solve this problem, showing competitive or even superior performance compared with supervised learning. The main idea is to learn representation from a large set of unlabeled images via pretext tasks rather than annotation. Research efforts have also been devoted to semi-supervised learning [44, 41, 65, 31], such as self training and consistency regularization, which collaboratively exploits both labeled data and large-scale unlabeled data to boost performance. Existing self-supervised and semi-supervised methods are mainly evaluated on ImageNet[8] and MSCOCO[37], where data labels are artificially removed for demonstration. There is no available benchmark for investigating advanced self-supervised and semi-supervised techniques for autonomous driving with real large-scale data.

To boost the development of real-world autonomous driving systems, we develop the first and largest-**S**cale **O**bject **D**etection benchmark for **A**utonomous driving (SODA10M) that contains 10 million road images. Our SODA10M dataset can be distinguished from existing datasets from three aspects, including *scale*, *diversity* and *generalization*.

*Scale*. As shown in Table 1, SODA10M is significantly larger than existing autonomous driving datasets like BDD100K [64] and Waymo [49]. It contains 10 million images of road scenes, which is ten times more than Waymo [49]. Specifically, 20K images with tightly fitting high-quality 2D bounding boxes while 10M images are unlabeled. All images contain detailed geographical, chronological and weather information.

*Diversity*. As shown in Fig. 1, SODA10M comprises images covering four seasons in 32 cities under different scenarios (e.g., urban, rural) and circumstances (e.g., night, rain, snow), while most present self-driving datasets [67, 64, 49] are less diverse. The changing scenarios and circumstances result in significant domain gaps in SODA10M. Specifically, the labeled training set contains only one domain, while the validation set and the unlabeled set contain 18 and 48 domains, respectively, which can serve as a challenging benchmark for unsupervised or semi-supervised domain adaptation.

*Generalization*. The largest scale and diversity ensure SODA10M's superior generalization ability as a pre-training dataset over all existing autonomous-driving datasets. Observed from evaluations of existing self-supervised algorithms, the representations learned from SODA10M unlabeled set are superior to that learned from other driving datasets like Waymo [49], i.e., 38.9% vs. 37.1% in mAP for object detection task on SODA10M labeled set and 75.2% vs. 73.8% for semantic segmentation task on Cityscapes [7] when using MoCov1 [21] (see Sec. 4.3 for more details).

Table 1: Comparison of dataset statistics with existing benchmarks. Night/Rain indicates whether the dataset has domain information related to night/rainy scenes. Video represents whether the dataset provides video format or detailed chronological information. Note that only 93K images of nuScenes are labeled with 2D format. SODA10M, which focuses on self/semi-supervised learning, contains 10M unlabeled and 20K labeled images.

| Dataset | Images | Cities | Night/Rain | Video | Categories | Boxes | Resolution |
|---------|--------|--------|------------|-------|------------|-------|------------|
| Caltech Pedestrian [10] | 249K | 5 | ✗/✗ | ✓ | 1 | 347K | 640×480 |
| KITTI [14] | 15K | 1 | ✗/✗ | ✗ | 3 | 80K | 1242×375 |
| Citypersons [67] | 5K | 27 | ✗/✗ | ✗ | 1 | 35K | 2048×1024 |
| BDD100K [64] | 100K | 4 | ✓/✓ | ✓ | 10 | 1.8M | 1280×720 |
| nuScenes [1] | 1.4M | 2 | ✓/✓ | ✓ | 23 | 0.8M | 1600×900 |
| Waymo Open [49] | 1M | 3 | ✓/✓ | ✓ | 3 | 9.9M | 1920×1280 |
| SODA10M (Ours) | **10M** | **32** | ✓/✓ | ✓ | 6 | 149K | 1920×1080 |

We provide experiments and in-depth analysis of existing supervised detection models, prevailing self-supervised and semi-supervised approaches on SODA10M. Observation can be made that simple self-supervised methods (e.g., MoCo-v1 [21]) achieve better results than the dense contrastive ones (e.g., DenseCL [56]) on SODA10M unlabeled set and semi-supervised methods work much better than self-supervised methods, even with a smaller set of unlabeled data (1-million vs. 5-million).

This benchmark will be used to hold the ICCV2021 SSLAD challenge, which aims to investigate current ways of building next-generation industry-level autonomous driving systems by resorting to self-supervised and semi-supervised learning. The SODA10M dataset and more up-to-date related information have been released and will be maintained weekly.

## 2 Related Work

**Driving datasets** have gained enormous attention due to the popularity of autonomous self-driving. Several datasets focus on detecting specific objects such as pedestrians [10, 67]. Cityscapes [7] provides instance segmentation on sampled frames, while BDD100K [64] is a diverse dataset under various weather conditions, time and scene types for multitask learning. For 3D tasks, KITTI Dataset [15, 14] was collected with multiple sensors, enabling 3D tasks such as 3D object detection and tracking. Waymo Open Dataset [49] provides large-scale annotated data with 2D and 3D bounding boxes, and nuScenes Dataset [1] provides rasterized maps of relevant areas.

**Supervised learning** methods for object detection can be roughly divided into single-stage and two-stage models. One-stage methods [36, 12, 38] directly outputs probabilities and bounding box coordinates for each coordinate in feature maps. On the other hand, two-stage methods [22, 45, 35] use a Region Proposal Network (RPN) to generate regions of interests, then each proposal is sent to obtain classification score and bounding-box regression offsets. By adding a sequence of heads trained with increasing IoU thresholds, Cascade RCNN [2] significantly improves detection performance. With the popularity of the vision transformer, more and more transformer-based object detectors [55, 40] have been proposed.

**Self-supervised learning** approaches can be mainly divided into pretext tasks [9, 66, 43, 42] and contrastive learning [21, 5, 4, 19]. Pretext tasks often adopt reconstruction-based loss functions [9, 43, 17] to learn visual representation, while contrastive learning is supposed to pull apart negative pairs and minimize distances between positive pairs, achieved by training objectives such as InfoNCE [53]. MoCo [21, 5] constructs a queue with a large number of negative samples and a moving-averaged encoder, while SimCLR [4] explores the composition of augmentations and the effectiveness of non-linear MLP heads. SwAV [3] introduces cluster assignment and swapped prediction to be more robust about false negatives, and BYOL [19] demonstrates that negative samples are not prerequisite to learn meaningful visual representation. For video representation learning, early methods are based on input reconstruction [24, 25, 32, 33], while others define different pretext tasks to perform self-supervision, such as frame order prediction [34], future prediction [48, 54] and spatial-temporal jigsaw [30]. More recently, contrastive learning is integrated to learn temporal changes [18, 63].

**Semi-supervised learning** methods mainly consist of self training [61, 59] and consistency regularization [46, 65, 20]. Consistency regularization tries to guide models to generate consistent predictions between original and augmented inputs. In the field of object detection, previous works focus on training detectors with a combination of labeled, weakly-labeled or unlabeled data [26, 50, 13], while recent works [28, 47] train detectors with a small set of labeled data and a larger amount

of unlabeled images. Specifically, STAC [47] pre-trains the object detector with labeled data and generate pseudo labels on unlabeled data, which are used to finetune the pre-trained model. Unbiased Teacher [39] further improves the process of generating pseudo labels via teacher-student mutual learning.

## 3 SODA10M

We collect and release a large-scale 2D dataset to promote significant progress of self-supervised and semi-supervised learning in autonomous driving. Our SODA10M contains 10M unlabeled images and 20K labeled images, which is split into training(5K), validation(5K) and testing(10K) sets.

### 3.1 Data Collection

The image collection task is distributed to the tens of thousands of taxi drivers in crowdsourcing. They have to use the mobile phone or driving recorder (1080P+) to obtain images every ten seconds per frame. Horizon needs to be kept at the center of the image, and the occlusion inside the car should not exceed 15% of the whole picture. Images should be obtained in diverse weather conditions, periods, locations and cities to achieve more diversity. After receiving each batch of images from the suppliers, a random 5% of pictures will be selected for manual verification. Batches of images with a pass rate below 95% will be returned for rectification.

**Driving hours**. The span of driving time for SODA10M (collected every ten seconds per frame) is 27833 hrs, which is much higher than the current large-scale datasets (5.5 hrs of nuScenes [1], 6.4 hrs of Waymo [49] and 1111.1 hrs of BDD100K [64]).

**Data Split.** We carefully select 5K training set, 5K validation set, 10K testing set with disjoint sequence id (same sequence id denotes the corresponding images are taken by same car on same day). Then we remove the images with same sequence id as the labeled set and randomly select 10-million images to construct the unlabeled part of SODA10M. Considering the convenience for downloading and using, we further divide 10-million unlabeled images into 10 splits by time sequence, with each split containing 1-million images.

**Data Protection.** The driving scenes are collected in permitted areas. We comply with the local regulations and avoid releasing any localization information, including GPS and cartographic information. For privacy protection, we actively detect any object on each image that may contain personal information, such as human faces and license plates, with a high recall rate. Then, we blur those detected objects to ensure that no personal information is disclosed. Detailed licenses, terms of use and privacy are listed in Appendix A.

### 3.2 Annotation

Image tags (i.e., weather conditions, location scenes, periods) for all images and 2D bounding boxes for labeled parts should be annotated for SODA10M. To ensure high quality and efficiency, the whole annotation progress can be divided into the following three different steps.

**Pre-annotation**: In order to ensure efficiency, a multi-task detection model, which is based on Faster RCNN [45] and searched backbone [29, 62], is trained on millions of Human-Vehicle images with bounding-box annotation and generate coarse labels for each image first.

**Annotation**: Based on pre-annotated labels, annotators keep the accurate ones and correct the inaccurate labels. Each image is distributed to different annotators, and the images with the same annotation will be passed to the following process; otherwise, they would be distributed again. All annotators must participate in several courses and pass the examination for standard labeling.

**Examination**: Senior annotators with rich annotation experience will review the image annotations in the second step, and the missing or incorrectly labeled images will be sent back for re-labeling.

We exhaustively annotated car, truck, pedestrian, tram, cyclist and tricycle with tightly-fitting 2D bounding boxes in 20K images. The bounding-box label is encoded as $(x, y, w, h)$, where $x$ and $y$ represent the top-left pixel of the box, and $w$ and $h$ represent the width and length of the box.

### 3.3 Statistics

**Labeled Set.** The labeled set contains 20K images with full annotation. There are 5K images for training, 5K images for validation and 10K images for testing. As shown in Fig. 2, the training set

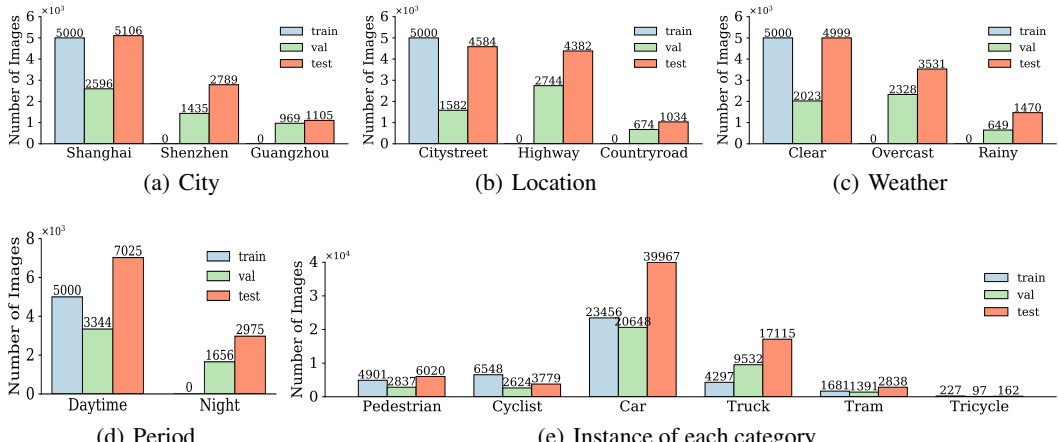

Figure 2: Statistics of the labeled set. (a) Number of images in each city. (b) Number of images in each location. (c) Number of images in each weather condition. (d) Number of images in each period. (e) Number of instances in each category.

only contains images obtained in city streets of Shanghai with clear weather in the daytime, while the validation and testing sets have three weather conditions, locations, cities and two different periods of the day. Considering the small gap between domains in different cities, we define 18 fine-grained domains through the pairwise combination of the remaining domains. The number of images in each fine-grained domain in the validation set and testing set are shown in Appendix D.

**Unlabeled Set.** The unlabeled set contains 10M images with diverse attributes. As shown in Fig. 3(a), the unlabeled images are collected among 32 cities, covering a large part of eastern China. Compared with the labeled set, the unlabeled set contains not only many more cities but also additional scenes such as residential, snowy and dawn/dusk, according to the gray part in Fig. 3(b), Fig. 3(c) and Fig. 3(d). The rich diversity in SODA10M unlabeled set ensures the generalization ability to transfer to other downstream autonomous driving tasks as a pre-training or self-training dataset.

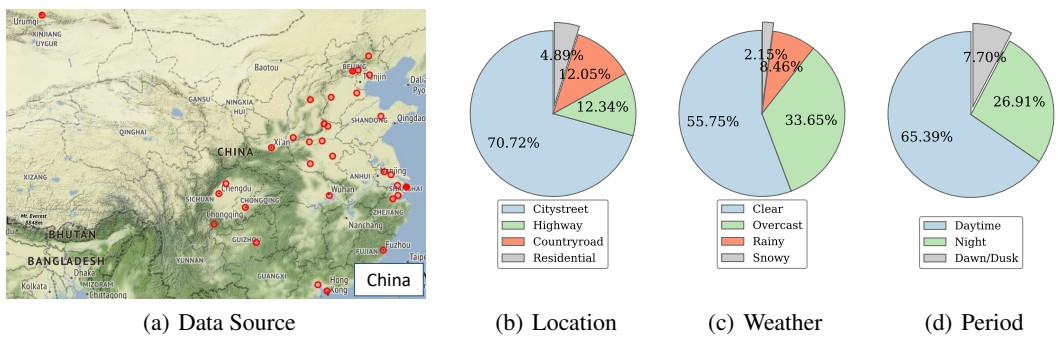

Figure 3: Statistics of the unlabeled set. (a) Geographical distribution of our data sources. SODA10M is collected from 32 cities, and darker color indicates greater quantity. (b) Number of images in each location. (c) Number of images in each weather condition. (d) Number of images in each period.

**Diversity Comparison.** We compare the diversity between SODA10M and other large-scale datasets (including nuScenes [1], Waymo [49] and BDD100K [64]) in period, weather and location fields. As shown in Table 2, our SODA10M is more diverse in all fields compared with nuScenes [1] and Waymo [49]. Although BDD100K [64] achieves competitive diversity with SODA10M in above three fields, equipped with more data (10M vs. 100K) and more cities where the data is collected from (32 vs. 4), SODA10M can better serve as the dataset and benchmark which focuses on solving self/semi-supervised learning problems.

Table 2: Diversity comparison between SODA10M and other datasets (i.e., nuScenes [1], Waymo [49] and BDD100K [64]), where '-' denotes for not having annotations in this field.

| Dataset | Period | Weather | Location |
|---|---|---|---|
| nuScenes [1] | Day: 88.3%, Night: 11.7% | Sunny: 80.4%, Rain: 19.6% | - |
| Waymo [49] | Day: 80.7%, Night: 9.8%, Dawn/Dusk: 9.5% | Sunny: 99.4%, Rain: 0.6% | - |
| BDD100K [64] | Daytime: 52.6%, Night: 40.1%, Dawn/Dusk: 7.3% | Clear: 60.6%, Overcast: 14.2%, Rainy: 8.1%, Snowy: 8.9%, Partly cloudy: 8.0%, Foggy: 0.2% | City street: 62.3%, Highway: 25.1%, Residential: 11.8%, Parking lot: 0.5%, Gas stations: 0.1%, Tunnel: 0.2% |
| SODA10M | Daytime: 65.4%, Night: 26.9%, Dawn/Dusk: 7.7% | Clear: 55.7%, Overcast: 33.6%, Rainy: 8.5%, Snowy: 2.2% | City street: 70.7%, Highway: 12.3%, Country road: 12.1%, Residential: 4.9% |

## 4  Benchmark

As SODA10M is regarded as a new autonomous driving benchmark, we provide the fully supervised baseline results based on several representative one-stage and two-stage detectors. With the massive amount of unlabeled data, we then study the generalization ability of state-of-the-art self-supervised and semi-supervised methods based on SODA10M and give insights into developing future models. Methods used for building this benchmark are representative samples of Fig. 4. To make the experiments easily reproducible, the code of all used methods has been open-sourced, and detailed experiment settings and training time comparisons are provided in Appendix B.

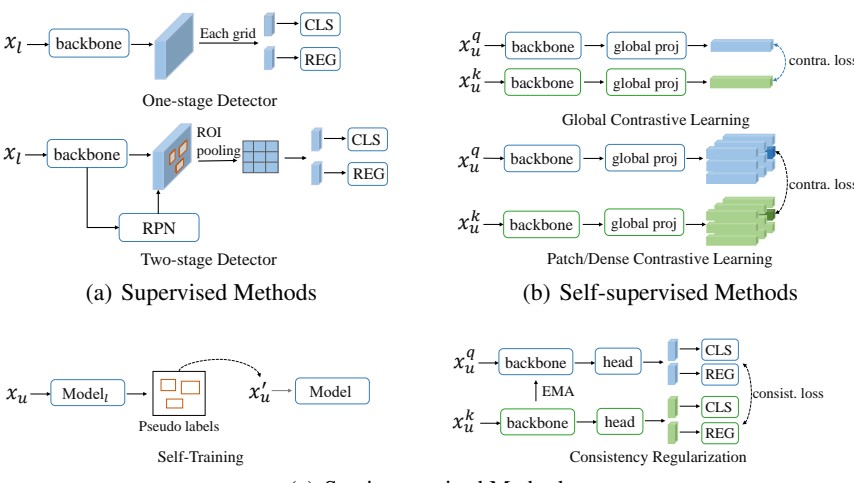

(a) Supervised Methods  (b) Self-supervised Methods

(c) Semi-supervised Methods

Figure 4: Overview of different methods used for building SODA10M benchmark. $X_l$ and $X_u$ denote for labeled set and unlabeled set. $q, k$ represent for different data augmentations. For semi-supervised learning methods, the labeled set is also involved in training progress with supervised loss.

### 4.1  Basic Settings

We utilize Detectron2 [57] as our codebase for the following experiments. Following the default settings in Detectron2, we train detectors with 8 Tesla V100 with a batch size 16. For the 1x schedule, the learning rate is set to 0.02, decreased by a factor of 10 at 8th, 11th epoch of total 12 epochs, while 2x indicates 24 epochs. Multi-scale training and SyncBN are adopted in the training process and precise-BN is used during the testing process. The image size in the testing process is set to $1920 \times 1080$. Unless specified, the algorithms are tested on the validation set of SODA10M. COCO API [37] is adopted to evaluate the detection performance for all categories.

### 4.2  Supervised Learning Benchmark

As shown in Table 3, the detection results of four popular object detectors (RetinaNet [36], Faster RCNN [45], Cascade RCNN [2] ) are compared. We observe that in the 1x schedule, Faster RCNN

exceeds RetinaNet in mAP by 5.3% with a larger number of parameters, which is consistent with the traditional difference of single-stage and two-stage detectors. Equipped with a stronger head, Cascaded RCNN can further surpass Faster RCNN by a large margin (3.9%). Observation can also be made that training with a longer schedule can further improve the performance.

Table 3: Detection results(%) of baseline models on SODA10M dataset.

| Model | Split | mAP | Pedestrian | Cyclist | Car | Truck | Tram | Tricycle | Params |
|---|---|---|---|---|---|---|---|---|---|
| RetinaNet [36] 1x | Val | 32.7 | 23.9 | 37.3 | 55.7 | 40.0 | 36.6 | 3.0 | 36.4M |
| RetinaNet [36] 2x | Val | 35.0 | 26.6 | 39.4 | 57.2 | 41.8 | 38.2 | 6.5 | 36.4M |
| RetinaNet [36] 2x | Test | 34.0 | 24.9 | 36.9 | 57.5 | 44.7 | 32.1 | 7.8 | 36.4M |
| Faster RCNN [45] 1x | Val | 37.9 | 31.0 | 43.2 | 58.3 | 43.2 | 41.3 | 10.5 | 41.4M |
| Faster RCNN [45] 2x | Val | 38.7 | 32.5 | 43.6 | 58.9 | 43.7 | 40.8 | 12.6 | 41.4M |
| Faster RCNN [45] 2x | Test | 36.7 | 29.5 | 40.1 | 59.7 | 47.2 | 32.3 | 11.7 | 41.4M |
| Cascade RCNN [2] 1x | Val | 41.9 | 34.6 | 46.7 | 61.9 | 47.2 | 45.1 | 16.0 | 69.2M |
| Cascade RCNN [2] 1x | Test | 39.4 | 31.9 | 43.4 | 62.6 | 50.0 | 36.8 | 11.9 | 69.2M |

Precision recall (PR) curves (from COCO eval API [37]) of each category for Faster RCNN 1x are shown in Fig. 5. Observation can be made that for categories with a small number of instances (Tricycle, Tram and Pedestrian), the error types are mainly from many false positives (FP) with class confusion, which is shown in the green part. On the contrary, for the primary category like Car, FP has little impact on the performance. Note that each category in SODA10M is a singleton supercategory so its Sim result is identical to Loc. We also illustrate the PR curves of Cascade RCNN 1x and find that the error type is basically consistent with Faster RCNN while Cascade RCNN shows stronger detection performance.

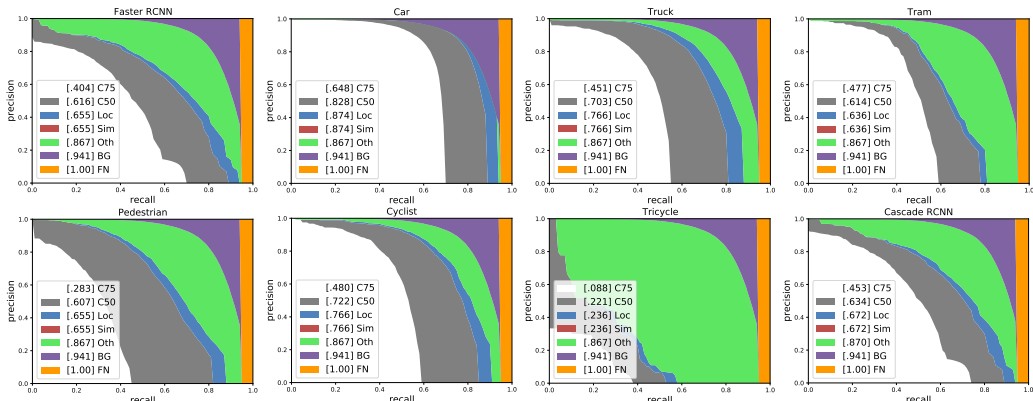

Figure 5: Precision recall curves of each category for Faster RCNN 1x and Cascaded RCNN 1x.

## 4.3 Self-Supervised Learning Benchmark

Self-supervised learning, especially contrastive learning methods, has raised attraction recently as it learns effective transferable representations via pretext tasks without semantic annotations. Traditional self-supervised algorithms [11, 42, 16] are usually pre-trained on ImageNet, while recent works [6, 51] have shown the consistency between upstream and downstream data distribution has a positive impact on the final performance. Therefore, we mainly compare the performance of existing mainstream self-supervised methods pre-trained on ImageNet and autonomous driving datasets, including SODA10M, BDD100K [64], nuScenes [1] and Waymo [49].

We follow the default settings in OpenSelfSup[1] to train six state-of-the-art standard self-supervised learning methods, including MoCo-v1 [21], MoCo-v2 [5], SimCLR [4], SwAV [3], DetCo [58], DenseCL [56], and evaluate their performance by fine-tuning the pre-trained models on the SODA10M labeled data and other self-driving datasets like BDD100K [64] and Cityscapes [7] to verify the

---

[1] https://github.com/open-mmlab/OpenSelfSup

Table 4: Detection results(%) of self-supervised models evaluated on SODA10M labeled set, Cityscapes [7] and BDD100K [64]. mIOU(C), mIOU(B) denotes for semantic segmentation performance on Cityscapes and BDD100K respectively. † represents for training with additional 5-million data. FCN-16s is a modified FCN with stride 16 used in MoCo [21]. 1x and 90k denote finetuning 12 epochs and 90k iterations, respectively.

| Pre-trained Dataset | Method | Faster-RCNN 1x | | | RetinaNet 1x | | | FCN-16s 90k | |
|---|---|---|---|---|---|---|---|---|---|
| | | mAP | AP50 | AP75 | mAP | AP50 | AP75 | mIOU (C) | mIOU (B) |
| | random init | 23.0 | 40.0 | 23.9 | 11.8 | 20.8 | 12.0 | 65.3 | 50.7 |
| | super. IN | 37.9 | 61.6 | 40.4 | 32.7 | 53.9 | 33.9 | 74.6 | 58.8 |
| ImageNet [8] | MoCo-v1 [21] | 39.0 | 62.0 | 41.6 | 33.8 | 54.9 | 35.2 | 75.3 | 59.7 |
| | MoCo-v2 [5] | 39.5 | 62.7 | 42.4 | 35.2 | 56.4 | 36.8 | 75.7 | 60.0 |
| | SimCLR [4] | 37.0 | 60.0 | 39.4 | 29.0 | 49.0 | 29.3 | 75.0 | 59.2 |
| | SwAV [3] | 35.7 | 59.9 | 36.9 | 26.4 | 45.7 | 26.3 | 73.0 | 57.1 |
| | DetCo [58] | 38.7 | 61.8 | 41.3 | 33.3 | 54.7 | 34.3 | 76.5 | 61.6 |
| | DenseCL [56] | 39.9 | 63.2 | 42.6 | 35.7 | 57.3 | 37.2 | 75.6 | 59.3 |
| BDD100K [64] | MoCo-v1 [21] | 37.1 | 60.1 | 39.2 | 31.1 | 51.6 | 32.1 | 74.5 | 57.9 |
| | MoCo-v2 [5] | 37.8 | 60.2 | 40.4 | 31.6 | 51.8 | 32.9 | 74.4 | 57.5 |
| nuScenes [1] | MoCo-v1 [21] | 36.2 | 58.9 | 38.1 | 29.3 | 49.2 | 29.9 | 73.6 | 57.0 |
| | MoCo-v2 [5] | 36.8 | 59.6 | 39.3 | 30.8 | 51.2 | 31.7 | 73.8 | 56.8 |
| Waymo [49] | MoCo-v1 [21] | 37.1 | 59.8 | 39.3 | 31.2 | 51.8 | 32.3 | 73.8 | 57.0 |
| | MoCo-v2 [5] | 37.1 | 59.7 | 39.4 | 31.4 | 52.0 | 32.4 | 73.5 | 56.6 |
| | DetCo [58] | 36.3 | 59.1 | 38.4 | 29.4 | 49.4 | 29.9 | 74.6 | 58.2 |
| SODA10M | MoCo-v1 [21] | 38.9 | 62.1 | 41.2 | 33.4 | 54.4 | 34.6 | 75.2 | 59.3 |
| | MoCo-v1† [21] | 39.0 | 62.6 | 41.9 | 33.8 | 55.2 | 35.2 | 75.5 | 59.5 |
| | MoCo-v2 [5] | 38.7 | 61.5 | 41.4 | 33.3 | 54.1 | 34.7 | 74.2 | 58.2 |
| | MoCo-v2† [5] | 38.6 | 61.3 | 41.4 | 33.2 | 54.6 | 34.6 | 74.5 | 58.9 |
| | SimCLR [4] | 35.9 | 59.5 | 37.4 | 28.7 | 48.7 | 29.1 | 73.3 | 57.3 |
| | SimCLR† [4] | 37.1 | 60.9 | 39.8 | 30.5 | 51.3 | 31.2 | 73.5 | 58.8 |
| | SwAV [3] | 33.4 | 57.1 | 34.5 | 24.5 | 43.2 | 24.6 | 68.6 | 54.2 |
| | DetCo [58] | 37.7 | 60.6 | 40.1 | 32.4 | 54.1 | 33.4 | 74.1 | 59.3 |
| | DenseCL [56] | 38.1 | 60.8 | 40.5 | 33.6 | 54.8 | 35.0 | 75.2 | 57.4 |
| | Video MoCo-v1 [21] | 34.9 | 57.8 | 36.6 | 27.9 | 47.3 | 28.2 | 73.6 | 57.3 |
| | Video MoCo-v2 [5] | 34.8 | 57.0 | 36.5 | 28.9 | 48.6 | 29.5 | 74.4 | 56.8 |
| | Video VINCE [18] | 34.9 | 57.7 | 36.9 | 27.6 | 47.1 | 28.0 | 72.6 | 57.4 |
| | Video VINCE+Jigsaw [18] | 35.5 | 58.1 | 37.0 | 28.2 | 48.1 | 28.6 | 74.1 | 56.9 |

generalization ability. For video-based self-supervised learning, MoCo-v1 [21], MoCo-v2 [5] and VINCE [18] are adopted. To ensure fairness, we apply the same data augmentation with VINCE to MoCo-v1 and MoCo-v2 to exploit temporal information and extra jigsaw augmentation to VINCE for better results. Due to the limit of hardware resources, we only use a 5-million unlabeled subset in each experiment by default, while we also make full use of the other 5-million subset in a sequential training manner, following Hu et al. [27]. Specifically, the model pre-trained on the first subset will be used as initialization to continue pre-training on the second one. We adopt 3700-epoch, 220-epoch, 325-epoch and 60-epoch pre-training on BDD100K [64], nuScenes [1] and Waymo [49] and SODA unlabeled set for image-based methods respectively, to maintain similar GPU hours with pre-training 200 epochs on ImageNet for fair comparison. Video-based approaches are trained for 800 epochs by considering time limit.

**ResNet-based Methods.** We pre-train on three different datasets (ImageNet, Waymo and SODA10M unlabeled set), and then report the transfer performance on three downstream tasks (detection on SODA10M labeled set, semantic segmentation on Cityscapes and BDD100K) in Table 4. For different downstream detection tasks listed in this table, the MoCo methods (MoCo-v1 [21], MoCo-v2 [5]) and dense contrastive methods (DenseCL [56], DetCo [58]) can achieve better results, while the other methods perform even worse than ImageNet fully supervised pre-train. We also observe dense contrastive methods show excellent results when pre-trained on ImageNet, but relatively poor on SODA10M unlabeled set. Experiments show that the model pre-trained on ImageNet performs equivalent or better than the one in SODA10M, which is because the existing self-supervised methods are often designed for simple scenes like ImageNet and fail to deal with the complex driving scene. By comparing the results of the same self-supervised algorithm on other autonomous driving datasets, we verify that the diversity of SODA10M data can bring better generalization ability. Besides, more pre-training iterations will bring better performance. The above results inspire us to design suitable self-supervised tasks or different pre-training strategies according to complex driving scenarios. At

the same time, the diversity of SODA10M unlabeled set can also ensure that SODA10M is a superior upstream pre-training dataset. More downstream tasks (e.g., object detection, instance segmentation) and comparisons on 2x schedule are illustrated in Appendix C.

**Video-based Methods.** Since our unlabeled set has detailed timing information for videos, we also transform the unlabeled set into video frames whose interval is 10 seconds and perform contrastive learning on these sequential frames. We use the same unlabeled set with 5-million images as ResNet-based models. After transformation, we get around 90K videos. We train several popular algorithms with ResNet-50 backbone, and results are shown at the bottom of Table 4. Since augmentations in MoCo-v1 and MoCo-v2 are the same as VINCE, their performances are close to each other. With the stronger augmentation jigsaw, VINCE performs better on Faster RCNN.

**Transformer-based Methods.** In addition to pre-training with the traditional ResNet [23] backbone, we also provide the self-supervised result of transformer-based backbone on SODA10M dataset. We choose PVT-small [55] as the backbone by considering the training efficiency and easy deployment on object detection tasks. Experiment results in Table 5 show that simply applying traditional self-supervised learning methods results in a small drop (about 1-3%) in performance compared with ImageNet supervised pre-training. These results inspire us when pre-training a transformer-based model under a self-supervised scheme, we need to develop some specific algorithms based on its special structure, such as DeiT [52] and Swin-SSL [60].

Table 5: Detection results(%) of self-supervised models evaluated on SODA10M labeled set, Cityscapes (C) and BDD100K (B) with Transformer model (PVT). All models are pre-trained on SODA10M unlabeled set.

| | PVT-small [55] 1x | | | PVT-small [55] 2x | | | PVT-small [55] 1x | | PVT-small [55] 90k | |
| Model | mAP | AP50 | AP75 | mAP | AP50 | AP75 | mAP-C | AP50-C | mIOU-C | mIOU-B |
|---|---|---|---|---|---|---|---|---|---|---|
| random init | 20.6 | 37.9 | 20.1 | 22.5 | 40.3 | 22.2 | 29.8 | 54.9 | 52.5 | 35.4 |
| super. IN | 33.8 | 57.3 | 35.2 | 33.0 | 55.1 | 33.9 | 33.8 | 60.0 | 60.0 | 41.8 |
| MoCo-v1 [21] | 28.7 | 50.3 | 29.1 | 28.5 | 49.7 | 29.1 | 30.4 | 56.4 | 59.2 | 40.3 |
| MoCo-v2 [5] | 26.2 | 46.8 | 26.3 | 26.7 | 46.5 | 27.4 | 28.3 | 52.5 | 58.1 | 39.4 |
| BYOL [19] | 27.4 | 49.2 | 26.9 | 26.9 | 47.4 | 27.1 | 28.0 | 53.1 | 57.6 | 40.5 |
| SimCLR [4] | 30.2 | 54.1 | 29.7 | 30.4 | 53.3 | 30.7 | 30.8 | 56.6 | 58.5 | 40.9 |

### 4.4 Semi-Supervised Learning Benchmark

Semi-supervised learning has also attracted much attention because of its effectiveness in utilizing unlabeled data. We compare the naive pseudo labeling method with present state-of-the-art semi-supervised methods for object detection (*i.e.*, STAC [47] and Unbiased Teacher [39]) on 1-million unlabeled images considering the time limit. Both methods achieve high performance with only 1-million unlabeled images. For pseudo labeling, we first train a supervised model on the training set with the ResNet-50 [23] backbone for 12 epochs. Then we predict results on the unlabeled set, a bounding box with a predicted score larger than 0.5 is selected as a predicted label. All semi-supervised methods exceed the results of using only labeled data. As for pseudo labeling, adding an appropriate amount of unlabeled data (50K to 100K) brings a greater improvement, but continuing to add unlabeled data (100K to 500K) results in a 1.4% decrease due to the larger noise. We follow the default settings in STAC and Unbiased Teacher, and change the input size to comply with SODA10M. Shown in in Table 6, the STAC exceeds pseudo labeling by 2.9%, and Unbiased Teacher continues to improve by 3.4% due to the combination of Exponential Moving Average (EMA) and Focal loss [36].

Table 6: Detection results(%) of semi-supervised models on SODA10M dataset. Pseudo labeling (50K), Pseudo labeling (100K) and pseudo labeling (500K) means using 50K, 100K and 500K unlabeled images, respectively.

| Model | mAP | AP50 | AP75 | Pedestrian | Cyclist | Car | Truck | Tram | Tricycle |
|---|---|---|---|---|---|---|---|---|---|
| Supervised | 37.9 | 61.6 | 40.4 | 31.0 | 43.2 | 58.3 | 43.2 | 41.3 | 10.5 |
| Pseudo Labeling (50K) | $39.3^{+1.4}$ | 61.9 | 42.4 | 32.6 | 44.3 | 60.4 | 43.8 | 42.4 | 12.1 |
| Pseudo Labeling (100K) | $39.9^{+2.0}$ | 62.7 | 42.6 | 33.1 | 45.2 | 60.7 | 44.8 | 43.3 | 12.1 |
| Pseudo Labeling (500K) | $38.5^{+0.6}$ | 61.0 | 41.3 | 32.1 | 43.4 | 59.6 | 42.6 | 42.2 | 11.0 |
| STAC [47] | $42.8^{+4.9}$ | 64.8 | 46.0 | 35.7 | 46.4 | 63.4 | 47.5 | 44.4 | 19.6 |
| Unbiased Teacher [39] | $46.2^{+8.3}$ | 70.1 | 50.2 | 33.8 | 50.2 | 67.9 | 53.9 | 55.2 | 16.4 |

Table 7: Detection results(%) in different domains on SODA10M dataset. IN indicates pre-trained on ImageNet, and SD means pre-trained on SODA10M unlabeled set. '-' means no validation image in this domain.

| Model | Overall mAP | City street (Car) | | | Highway (Car) | | | Country road (Car) | | |
|---|---|---|---|---|---|---|---|---|---|---|
| | | Clear | Overcast | Rainy | Clear | Overcast | Rainy | Clear | Overcast | Rainy |
| Daytime | | | | | | | | | | |
| Supervised | 43.1 | 70.0 | 64.9 | 56.6 | 68.3 | 65.9 | 65.9 | 69.4 | 63.5 | - |
| MoCo-v1 [21] IN | $44.2^{+1.1}$ | 71.5 | 65.8 | 56.9 | 69.0 | 66.8 | 67.3 | 72.0 | 66.0 | - |
| MoCo-v1 [21] SD | $43.8^{+0.7}$ | 71.3 | 66.0 | 55.8 | 69.4 | 67.4 | 68.0 | 72.8 | 65.5 | - |
| STAC [47] | $45.3^{+2.2}$ | 74.2 | 69.6 | 58.0 | 71.7 | 70.3 | 70.7 | 75.2 | 69.8 | - |
| Unbiased Teacher [39] | $47.7^{+4.6}$ | 73.0 | 68.1 | 55.3 | 69.1 | 62.0 | 71.3 | 72.6 | 70.0 | - |
| Night | | | | | | | | | | |
| Supervised | 21.1 | 36.3 | 37.7 | - | 37.5 | 37.3 | 79.5 | 38.9 | 72.8 | - |
| MoCo-v1 [21] IN | $22.0^{+0.9}$ | 39.5 | 43.4 | - | 41.7 | 41.5 | 80.6 | 42.5 | 73.2 | - |
| MoCo-v1 [21] SD | $22.7^{+1.6}$ | 41.6 | 46.2 | - | 42.1 | 41.8 | 79.8 | 45.4 | 74.1 | - |
| STAC [47] | $28.2^{+7.1}$ | 45.5 | 46.8 | - | 46.2 | 45.6 | 83.7 | 47.2 | 75.4 | - |
| Unbiased Teacher [39] | $39.7^{+18.6}$ | 65.3 | 66.2 | - | 66.2 | 67.2 | 83.6 | 67.5 | 75.2 | - |

## 4.5 Discussion

We directly compare the performance of state-of-the-art semi/self-supervised object detection methods with supervised Faster-RCNN in Table 7. In this table, we illustrate the overall mAP for daytime/night domain and car detection results of 18 fine-grained domains consisting of different periods, locations and weather conditions.

Observation can be made that there exists a huge gap between the domain of daytime and night. Since the supervised method is only trained on the data during the daytime, the gap between day and night is particularly obvious. By adding diverse unlabeled data into training, the self/semi-supervised methods show a more significant improvement in the night domain. Specifically for semi-supervised learning, Unbiased teacher [39] surpasses STAC [47] by a large margin in the night domain because it can address pseudo-labeling bias issues caused by class imbalance existing in ground-truth labels and the overfitting issue caused by the scarcity of labeled data. Besides, semi-supervised methods work much better than self-supervised methods either from the aspect of overall performance or the training time (2.8×8 GPU days vs. 8.4×8 GPU days for Unbiased teacher and MoCov1 respectively in Appendix B).

Inspired by the above results, we summarize some guidance dealing with SODA10M dataset. For self-supervised learning, different from ImageNet pre-training, simple methods (e.g., MoCov1 [21]) achieve better results than the dense contrastive methods (e.g., DenseCL [56]) on SODA10M unlabeled set. Concentrating on driving scenes, semi-supervised methods work much better than self-supervised methods when finetuning on SODA10M labeled set, even with a smaller set of unlabeled data (1-million vs. 5-million). Better performance will be achieved when combining self-supervised and semi-supervised methods. For both self and semi-supervised learning, model architecture design and efficient training will be promising topics on SODA10M for future research.

# 5 Conclusion

Focusing on self-supervised and semi-supervised learning, we present SODA10M, a large-scale 2D autonomous driving dataset that provides a small set of high-quality labeled data and a large amount of unlabeled data collected from various cities under diverse weather conditions, periods and location scenes. Comparing with the existing self-driving datasets, SODA10M is 10x larger than the largest dataset available Waymo and obtained in much more diversity. Furthermore, we build a benchmark for supervised, self-supervised and semi-supervised learning in autonomous driving and show that SODA10M can serve as a promising dataset for training and evaluating different self/semi-supervised learning methods. We hope that SODA10M can promote the exploration and standardized evaluation of advanced techniques for robust and real-world autonomous driving systems.

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
