# OpenReview forum: "SODA10M: Towards Large-Scale Object Detection Benchmark for Autonomous Driving"
_NeurIPS.cc/2021/Track/Datasets_and_Benchmarks/Round1 — Submitted to NeurIPS 2021 Datasets and Benchmarks Track (Round 1)_

### Official Review · Reviewer_MRJ9 · 2021-06-22
**A larger-scale dataset for 2D object detection in autonomous driving**

**Rating:** 6
**Confidence:** 3
**Correctness:** The datasets are constructed in a sou…
**Clarity:** This paper is well written.

**Strengths:**

+ A large-scale data with 10 million unlabeled images and 20K labeled images is presented. This dataset could be used to facilitate future research on autonomous driving.
+ The dataset is released tp public, which is nice.
+ Extensive experiments with many existing methods are performed, and authors also conduct in-depth analysis. This is important as it can provide baseline comparison for future research.
+ This paper is well written and organized.

**Weaknesses:**

- I think the comparison in Tab. 1 is not fair. Because the most images in SODA10M are unlabeled, and collecting these unlabeled are actually not that hard. It is necessary to point it out in Tab. 1.
- In table, it is better to point out the aims of different datasets. For example, the proposed SODA10M is aimed at supervised, self- and semi-supervised object detection.
- In abstract, the authors claimed that "some insights about how to develop future models" is provided. However, I didn't see in-depth analysis about this claim.
- Since one of the major goal is for unsupervised learning for object detection, why don't develop your own baseline model?

**Additional Feedback:**

Will you release the models used in experiments in this paper?

**Documentation:**

Documentation is clear.

**Ethics:**

I don't see any problems.

**Relation To Prior Work:**

Discussion with related works is sufficient and clear.

**Summary And Contributions:**

Addressing the problems of requiring extensive annotated data in existing dataset, this paper contributes a large-scale 2D object detection dataset called SODA10M. In specific, SODA10M contains 10 million unlabeled images and 0K labeled images. These images are collected from 32 different cities under different weather conditions, period and location scenes, demonstrating its diversity. Moreover, the authors provide extensive experiments and analyses of various existing methods.

---

> ### Author Response · Authors · 2021-07-09
> **Reply to Reviewer3**
>
> Thank you for your comments. All comments are summarized and addressed as follows.
>
> **Q1: Point out SODA10M construction details in Table 1.**
>
> The SODA10M construction details are added in the caption of Table 1.
>
> **Q2: Point out the aims of SODA10M and different datasets.**
>
> SODA10M is the first dataset aiming 2D self-supervised and semi-supervised learning object detection for autonomous driving, which is different from the current fully-labeled datasets focusing on supervised learning.
> We emphasize the aim of SODA10M at the Table 1 and the front of the conclusion in revised version.
>
> **Q3: Insights about how to develop future models.**
>
> The insights of developing models on SODA10M are already concluded in the last paragraph in Section 4.5 of previous version, including comparison of performance between self-supervised methods and semi-supervised methods, and some promising topics on SODA10M for future research, etc.
>
> **Q4: Develop our own methods.**
>
> This paper simply aims to release the dataset and benchmark based on existing popular and open-sourced methods, thus we do not propose our own baseline model.
> But we are working on SODA10M dataset to develop some new self/semi-supervised methods.
>
> **Others.**
>
> We plan to release the models used in experiments on the SODA10M website soon.

---

> > ### Comment · Reviewer_MRJ9 · 2021-07-19
> > **Reply to authors' reply**
> >
> > Thank you very much for your efforts in addressing my concerns. Overall, I think this work can bring some interest to the community. I don't have further questions. Since I am not an expert in this field. I will keep my score unchanged.

---

### Official Review · Reviewer_sJ63 · 2021-07-04
**Building 2D RGB detection benchmark for autonomous driving**

**Rating:** 4
**Confidence:** 4
**Correctness:** Please see the Weaknesses section.
**Clarity:** Yes. The paper is clear.

**Strengths:**

- SODA contains data from different geographical locations (32 cities), weathers and times.
- The paper shows that RGB detectors (Faster RCNN and RetinaNet) pre-trained on SODA dataset perform better compared to Waymo on SODA10M labelled dataset.
- FCNs trained on SODA10m transfer better compared to the Waymo for semantic segmentation.
- The paper also builds a benchmark for self-supervised and semi-supervised learning in autonomous driving which is super nice.

**Weaknesses:**

- SODA has six categories, much less than the nuScenes dataset, which contains 23. Even in evaluation, nuScenes [1] uses ten categories (way more than SODA).

- Although SODA has more total images, 2D/3D annotations are less than that of nuScenes [1], BDD100k [2], and Waymo. The number of boxes for SODA is only 149k, while the number of 2D boxes for nuScenes, BDD100k, and Waymo are 1.4m, 4.2m, and 12m, respectively. Thus, the number of boxes in SODA is orders of magnitude smaller than other released datasets.

- Comparison of diversity missing. It is worth comparing the diversity of SODA, nuScenes, Waymo, and BDD100k to get a sense of how SODA is more diverse than other datasets. In other words, a histogram plot of weather, times of the day, and areas comparing the four datasets similar to Figure 2 (but comparing with the statistics of other datasets) would be nice.

- Faster RCNN and RetinaNet pre-trained on the SODA dataset perform better than pre-training with Waymo on the SODA10M labeled dataset. However, this does not show that the SODA dataset is better compared to Waymo. The detectors pre-trained on SODA10M do not face a domain shift, while the detectors trained on Waymo face the issue of domain adaptation. Therefore, a more fair way to compare the performance is to train these detectors and evaluate a different dataset such as nuScenes. A more thorough evaluation is to train and test on all combinations of source/target datasets similar to Table 2 of [3] (Note that [3] does for 3D object detection and not for 2D object detection.)

- Please also benchmark the Table 3 performance against nuScenes [1] and BDD100k, which are more diverse compared to Waymo open.

- Comparison of driving hours with the current datasets.

- The supervised benchmark of SODA is small compared to other benchmarks. nuScenes has 40k images, Waymo Open has 200k images, while SODA only has 20k. Although I agree that the total number of images is more compared to nuScenes and Waymo, I doubt that the supervised learning community would use the SODA dataset for benchmarking their models.

- One of the major drawbacks of this dataset is that it only builds a 2D object detection benchmark. However, in general, the 2D RGB object detection benchmark is not enough. A dataset should incorporate the Lidar data (maybe Radar), maps, and 3D boxes for evaluation. The vision and machine learning community does not use one dataset for the task of 2D object detection and another dataset for carrying multi-sensor fusion or even 3D object detection. Do the authors plan to extend the dataset to include multi-sensor information as well as 3D detection benchmark?

References- [1] nuscenes: A multimodal dataset for autonomous driving, Caeser et al, CVPR 2020.
                      [2] BDD100K, Yu et al , CVPR 2020.
                      [3] Train in Germany, Test in USA, Wang et al, CVPR 2020.

**Additional Feedback:**

- In Implementation Details (Section B of the Appendix), please provide the details of data augmentation, learning rate policy and optimizer of each of the methods. It is also worthwhile to put the data split of each of the benchmark for clarity.

- KITTI resolution in Table 1 should be 1242 x 375

**Documentation:**

Yes, these are all available in the main paper and appendix.

**Ethics:**

I do not think their are ethical concerns. The authors do not release any location information, including GPS and cartographic information. They also blur out human faces and license plates using a high recall rate algorithm to avoid divulging personal information. Moreover, the authors bear all responsibility in case of violation of rights as per their appendix.

**Relation To Prior Work:**

The paper does not compare with nuScenes. The experimental evaluation also does not use BDD100k and nuScenes dataset.

**Summary And Contributions:**

The paper introduces the SODA dataset for 2D object detection, containing 10m unlabeled images and 20k labeled RGB images. The authors collect the dataset across different geographical locations, weather and times. The paper shows that RGB detectors (Faster RCNN and RetinaNet) pre-trained on the SODA dataset perform better than Waymo on the SODA10M labeled dataset. Moreover, FCNs trained on SODA10m transfer better compared to the Waymo for semantic segmentation. The paper also builds a benchmark for self-supervised and semi-supervised learning in autonomous driving.

Update :
Additional experiments (with different upstream datasets like nuScenes and BDD100K, finetuning on object detection tasks) shows that the SODA10M dataset performs better on object detection compared to current datasets. However, diversity comparisons in Table 2 shows that the SODA10M dataset is less diverse compared to BDD100k based on period/weather/location. Therefore, I am still not convinced that the dataset is super useful to the community.

---

> ### Author Response · Authors · 2021-07-09
> **Reply to Reviewer2**
>
> Thanks for your comments and questions about SODA10M. All comments are summarized and addressed as follows.
>
> **Q1: The number of the labeled images and annotations is limited.**
>
> Here, we explain it from three different aspects:
>
> Firstly, SODA10M focuses on benchmarking self-supervised and semi-supervised 2D object detection instead of building a supervised benchmark. Thus we collect 10 million unlabeled images (largest so far) with great diversity and sufficient 20K labeled data for evaluation.
> Since SODA10M has more annotations than well-known datasets like KITTI or PASCAL VOC, we believe that the number of labeled images in SODA10M is sufficient.
>
> Secondly, in addition to the number of labeled images, annotation frequency is another important factor.
> It's worth mentioning that SODA10M is annotated at the sampling frequency of every ten seconds per frame, which means it covers diverse or efficient samples to train the detector.
> Although Waymo provides 9.9 million bounding boxes, with an annotation frequency of ten frames per second, most of the adjacent samples are similar and redundant.
>
> Finally, we plan to release more labeled images (i.e. 100k) to meet the needs of the supervised learning community after the corresponding ICCV 2021 competition.
>
> **Q2: Not enough categories vs. nuScenes.**
>
> SODA10M contains six necessary classes, which cover the most appearing instances in driving scene.
> Besides, we note that well-known datasets like KITTI only has 3 classes (Car/Pedestrian/Cyclist) and Waymo also only contains 3 classes (Vehicle/Pedestrian/Cyclist).
> Although nuScenes has more categories, most categories are covered by SODA10M, e.g., adult, child, construction worker and police officer in nuScenes are indicated as Pedestrian in SODA10M.
>
> **Q3: Diversity comparison between different datasets.**
>
> The diversity comparison of different large-scale datasets (Waymo, nuScenes and SODA10M) is added to Section 3.3.
> From the Table 2 in Section 3.3, we find that SODA10M is more diverse in all fields (including period, weather and location) compared with nuScenes and Waymo datasets.
>
> **Q4: Report the self-supervised performance against other datasets like nuScenes and BDD100K.**
>
> Comparison of self-supervised pre-training on other upstream datasets (nuScenes \& BDD100K) will be updated to Table 4 in Section 4.3 by the end of discussion phase.
>
> **Q5: Comparison of driving hours.**
>
> The comparison of training hours between different datasets is added to Section 3.1.
> The driving time for SODA10M is 27833 hrs, which is much higher than the current datasets (5.5 hrs of nuScenes, 6.4 hrs of Waymo and 1111.1 hrs of BDD100K).
>
> **Q6: Domain shift when pre-training on Waymo and finetuning on SODA10M dataset**
>
> We already performed experiments of finetuning on BDD100K and Cityscapes datasets in the right part of Table 4 in previous version.
> The results show that detectors pre-trained on SODA10M are usually better compared with Waymo (with an average of 1.1\% improvement).
>
> **Q7: Without lidar data.**
>
> Due to the demand for great diversity and large-scale, the image collection task is distributed to the drivers through the crowdsourcing way.
> The car, without either lidar sensor or legal qualification of collecting 3D information (required by Chinese government), can not provide the additional point cloud information.
> Besides, the diving time of SODA10M is 27834 hrs refer to Q5, which makes it almost impossible to be distributed to the limited number of cars with lidar sensor equipped.
> Note that autonomous driving datasets like BDD100K doesn't contain 3D point cloud information either.
>
> **Others.**
>
> 1. Data split details are added to Section 3.1 and data split of each benchmark is shown in Appendix B of previous version and can be downloaded separately on the official website.
>
> 2. Thanks for your suggestions, the additional experimental details of each method and comparison with nuScenes are added to Appendix B and Table 1 in Section 1, respectively.
>
> 3. Thanks for your correctness, the image size of the KITTI dataset in Table 1 is fixed.

---

> > ### Comment · Reviewer_sJ63 · 2021-07-10
> > **Reply to authors' response**
> >
> > Thank you, authors, for your response and for adding all the relevant information to the paper. I appreciate the authors' comparisons on driving hours and benchmarking self-supervised performance against other datasets like nuScenes and BDD100K (which they will add later). I also thank them for putting the details of data augmentation, learning rate policy and optimizer in the appendix.
> >
> > **Result better on Object Detection due to Domain Shift**
> >
> > SODA is a 2D RGB object detection dataset and not intended for semantic segmentation. Finetuning on BDD100K and Cityscapes in Table 4 shows that detectors pre-trained on SODA10M are better than Waymo on semantic segmentation and NOT on object detection. I believe this is the biggest weakness of the paper as SODA does not benchmark its performance on the object detection benchmark with competitive datasets in a fair manner on all combinations of train and test.
> >
> > Therefore, I strongly advise the authors to compare the performance of SODA against Waymo, BDD100k, and nuScenes on the object detection benchmark on all combinations of train and test. BDD100k and nuScenes are more diverse than Waymo. My hunch is that BDD100k and nuScenes should be a stronger baseline compared to Waymo.
> >
> > Without these benchmarking results, a reader will not be convinced that the SODA is a better dataset than BDD100k, Waymo, and nuScenes on the object detection benchmark.
> >
> > **The number of the labeled images and annotations is limited.**
> >
> > Comparison with KITTI or Pascal VOC is not correct as these datasets are old. Therefore, I believe the comparison should be with more recent datasets such as BDD100k, nuScenes, and Waymo Open.
> > Let us divide total boxes by fps. Total frames/ fps for Waymo is ~1m boxes, while for SODA10m is 149k. Please compare the total boxes and fps for nuScenes and BDD100k as well.
> >
> > **Diversity Comparison**
> >
> > I appreciate the comparisons and numbers made explicit in Table 2 for Waymo and nuScenes. BDD100k is also a strong competitor to the SODA dataset. Please add the diversity comparison with BDD100k as well in Table 2.
> >
> > **Without Lidar**
> >
> > Although I agree that getting lidar data from crowdsourcing is difficult and expensive, and even BDD100k does not have lidar information, SODA will be less widely used compared to nuScenes and Waymo because it does not have lidar or other information.

---

> > > ### Author Response · Authors · 2021-07-13
> > > **Reply to Reviewer2's response**
> > >
> > > Thanks for your timely response and advice about SODA10M dataset. Your further comments are summarized and addressed as follows.
> > >
> > > **Q1: Additional experiments (with different upstream datasets like nuScenes and BDD100K, finetuning on object detection tasks).**
> > >
> > > According to your advice, we add finetuning experiments with different self-supervised methods on object detection task (BDD100K dataset) and instance segmentation task (Cityscapes dataset) in Table 5 of Appendix C.  Self-supervised performance against other datasets (nuScenes \& BDD100K) is also updated to Table 4 of Section 4.3 and Table 5 of Appendix C.
> > >
> > > The results are summarized as follows, where OD, IS, SS denote for finetuning with object detection task, instance segmentation task and semantic segmentation task respectively:
> > >
> > > | Pre-train| Method  | OD-BDD(mAP) | OD-SODA(mAP) | IS-Cityscape(mAP) | SS-BDD(mIOU) | SS-Cityscape(mIOU) |
> > > | :---  |  :---  |  :--:  | :--:  |  :--:  |  :--:  | :--:  |
> > > | BDD100K | MoCo-v1 |31.4  | 37.1 | 31.8 | 57.9 | 74.5 |
> > > |  | MoCo-v2 | 31.3 | 37.8 | 32.0 | 57.5 | **74.4** |
> > > | nuScenes | MoCo-v1 | 31.1 | 36.2 | 31.4 | 57.0 | 73.6 |
> > > |  | MoCo-v2 | 30.9 | 36.8 | 31.5 | 56.8 |73.8  |
> > > | Waymo | MoCo-v1 | 31.2 | 37.1 | 31.4 | 57.0 | 73.8 |
> > > | | MoCo-v2 | 31.1 | 37.1 | 31.8 | 56.6         | 73.5 |
> > > | SODA10M | MoCo-v1 | **31.5** | **38.9** | **33.9** | **59.3** | **75.2** |
> > > |  | MoCo-v2 | **31.4** | **38.7** | **33.7** | **58.2** | 74.2 |
> > >
> > > From the above table, we observe that detectors pre-trained on SODA10M are usually better than pretrained on BDD100K, nuScenes and Waymo dataset on object detection task (BDD100K, SODA10M), instance segmentation task (Cityscape) and semantic segmentation task (BDD100K, Cityscape).
> > >
> > > **Q2: The number of the labeled images and annotations is limited.**
> > >
> > > We agree  that our SODA has limited annotated images comparing to some other benchmarks like Waymo. However, we want to make some clarifications again:
> > >
> > > 1. SODA10M focuses on benchmarking self-supervised and semi-supervised 2D object detection instead of building a supervised benchmark which contains more annotations.
> > >
> > > 2. By comparison with KITTI or Pascal VOC, we show that SODA10M provides sufficient labeled samples to evaluate different methods.
> > >
> > > 3. We plan to provide more labeled images (about 100K) after the ongoing ICCV 2021 competition.
> > >
> > > We also provide the total boxes and fps information of each dataset as required.
> > > The bounding-box number for Waymo (10Hz), nuScenes (2Hz) and SODA10M (0.1Hz, ten seconds per frame) is 9.9M, 0.8M and 149k respectively, thus the "total boxes by fps" metric mentioned is 0.99M/Hz for Waymo, 0.4M/Hz for nuScenes and 1.49M/Hz for SODA10M.
> > > We don't compare with BDD100K because the fps information is unknown (each image is sampled from individual video).
> > >
> > > **Q3: Diversity comparison vs. BDD100K.**
> > >
> > > Diversity comparison with SODA10M and BDD100k is added to Table 2 in Section 3.3.
> > > SODA10M contains more data (10M vs. 100K) and more cities where the data is collected from (32 vs. 4) compared with BDD100K, which makes it better serve as the dataset and benchmark which focuses on solving self-supervised and semi-supervised learning problems.
> > >
> > > **Q4: Without lidar.**
> > >
> > > SODA10M is built to focus on 2D self/semi-supervised learning for autonomous driving system, 3D lidar information is not necessary so we don't provide this data.
> > >
> > > The 2D Challenge of ICCV2021 SSLAD Workshop based on SODA10M dataset already starts on CodaLab and attracts 30+ teams in the first week of total 3-month competition.
> > > We believe that as the biggest and the first autonomous driving dataset focusing on 2D self/semi-supervised learning, SODA10M can help the research community to boost the development of the real-world autonomous driving systems.

---

### Official Review · Reviewer_nRC5 · 2021-07-05

**Rating:** 7
**Confidence:** 2
**Correctness:** The content appears to correct.

**Strengths:**

1.	The datasets are much larger than previous ones in this domain and are more diverse. It could facilitate future work in autonomous driving.
2.	The datasets will be open-sourced and will be maintained by the authors.
3.	The authors give a detailed analysis of the statistics of the datasets and how the dataset is annotated. The authors also present benchmark results on the dataset, which could serve as the baselines in future work.


**Weaknesses:**

Only 20k images are annotated in the dataset. This could limit the usability of the datasets. The labeled and unlabeled parts could also cover different domains. It would be hard to evaluate the domains that are not annotated.

**Additional Feedback:**

NaN

**Clarity:**

The paper is clearly written, with detailed descriptions of the datasets, annotation, and benchmark results.

**Documentation:**

The webpage of this dataset provides sufficient instructions on how to download and make use of the dataset.

**Ethics:**

No ethical concern.

**Relation To Prior Work:**

Compared with the previous datasets, SODA10M is larger in size and is more diverse. However, only 20K of them are labeled.

**Summary And Contributions:**

This paper presents a large-scale dataset for object detection in autonomous driving systems. The data is collected from some major cities and provinces in China. The data covers different locations, such as city streets, highways, and country roads. It also covers different weather conditions, and both daytime and night. The authors promised to maintain the dataset in the future and the dataset will also be used in an ICCV challenge.

---

> ### Author Response · Authors · 2021-07-09
> **Reply to Reviewer1**
>
> We appreciate for your comments. All comments are summarized and addressed as follows.
>
> **Q1: The number of the labeled images is limited, and annotations do not cover all domains.**
>
> Here, we explain it from three different aspects:
>
> Firstly, SODA10M focuses on benchmarking self-supervised and semi-supervised 2D object detection instead of building a supervised benchmark. Thus we collect 10 million unlabeled images (largest so far) with great diversity and sufficient 20K labeled data for evaluation.
> Since SODA10M has more annotations than well-known datasets like KITTI or PASCAL VOC, we believe that the number of labeled images in SODA10M is sufficient.
>
> Secondly, in addition to the number of labeled images, annotation frequency is another important factor.
> It's worth mentioning that SODA10M is annotated at the sampling frequency of every ten seconds per frame, which means it covers diverse or efficient samples to train the detector.
> Although Waymo provides 9.9 million bounding boxes, with an annotation frequency of ten frames per second, most of the adjacent samples are similar and redundant.
>
> Finally, we plan to release more labeled images (i.e. 100k) to meet the needs of the supervised learning community after the corresponding ICCV 2021 competition.

---

> > ### Comment · Reviewer_nRC5 · 2021-07-13
> > **Thank you for the response**
> >
> > Thank you for the detailed response. I have also looked through the reviews of the other reviewers. Overall, I will keep my score. From the view of the dataset itself, I think it is a non-trial effort to collect such a dataset and I appreciate the open-source efforts. The dataset seems to be easily accessible, will be used to hold a data challenge, and will be maintained by the authors. From this aspect, I think this is a valuable contribution.
> >
> > However, I am not an expert in this domain and am not familiar with the SOTA algorithms in this domain. Thus, I will keep my confidence score as it is.

---

### Author Response · Authors · 2021-07-09
**Updates on the revised version**

Based on the reviewer's comments, here we provide the summary about the updates of the revised version, note that all updates are highlighted by orange color:

1. More details (e.g., comparison with nuScenes, explanation of SODA10M construction details) are added to Table 1 in Section 1. (R2, R3)

2. Comparison of driving hours and data split details are added to Section 3.1. (R2)

3. Detailed settings about data augmentation, learning rate policy and optimizer of each method are added to the Appendix B. (R2)

4. Diversity comparison between SODA10M and large-scale datasets (Waymo and nuScenes)  is added to the Section 3.3. (R2)

5. Comparison of self-supervised pre-training on other upstream datasets (nuScenes \& BDD100K) will be updated to Table 4 in Section 4.3 by the end of the discussion phase. (R2)

6. The aim of developing SODA10M dataset is further emphasized in the Conclusion (Section 5). (R3)

7. We plan to release more labeled images (100k) after the corresponding ICCV competition. (R1, R2)

Links related to SODA10M:

1. Dataset URL: [https://soda-2d.github.io](https://soda-2d.github.io)
2. Challenge URL: [https://competitions.codalab.org/competitions/33288](https://competitions.codalab.org/competitions/33288)
3. Workshop URL: [https://sslad2021.github.io](https://sslad2021.github.io/)

---

### Decision · Program_Chairs · 2021-07-26

**Decision:**

Reject

**Comment:**

This paper receives conflicting reviews. The AC committee spent quite some time discussing the merits and drawbacks of the paper. The main concern is about the paper's claims and arguments. The major drawback is the lack of strong support for this dataset to be a unique benchmark for self-supervised and semi-supervised approaches. The existing quantitative results don't show distinct findings of self-supervised learning methods on the proposed dataset. Even though the experiments on different datasets show that the pre-trained model can benefit from the new dataset, the improvement can hardly be claimed to be significant. The claim of more cities is weak, due to different definitions of cities in different countries. In addition, nuScenes and BDD100K have more unlabeled images than those used in the experiments. The paper doesn't discuss the effects of those unlabeled images in the other datasets.

However, we do encourage the authors to conduct further investigation into the proposed dataset. Besides the study of uniqueness for self-supervised learning, the images from Chinese cities may be a good complement to the existing datasets and there may be interesting findings in domain adaptation study.